# Influence of Professional Affiliation on Expert’s View on Welfare Measures

**DOI:** 10.3390/ani7110085

**Published:** 2017-11-15

**Authors:** Nina Dam Otten, Tine Rousing, Björn Forkman

**Affiliations:** 1Section for Animal Welfare and Disease Control, Department of Veterinary and Medical Sciences, Faculty of Health and Medical Sciences, University of Copenhagen, 1870 Frederiksberg C, Denmark; bjf@sund.ku.dk; 2Epidemiology and Management, Department of Animal Sciences, Aarhus University, 8830 Tjele, Denmark; tine.rousing@anis.au.dk

**Keywords:** animal welfare, expert opinion, stakeholders

## Abstract

**Simple Summary:**

Animal welfare can be assessed from different ethical points of view, which may vary from one individual to another. This is often met by including different stakeholders’ opinions in the process of adding up welfare benefits and or welfare risks. However, in order to obtain the most reliable results, these expert panels should be balanced; since experts’ professional affiliations can influence their judgment on different welfare aspects as shown in the present study.

**Abstract:**

The present study seeks to investigate the influence of expert affiliation in the weighing procedures within animal welfare assessments. Experts are often gathered with different backgrounds with differing approaches to animal welfare posing a potential pitfall if affiliation groups are not balanced in numbers of experts. At two time points (2012 and 2016), dairy cattle and swine experts from four different stakeholder groups, namely researchers (RES), production advisors (CONS), practicing veterinarians (VET) and animal welfare control officers (AWC) were asked to weigh eight different welfare criteria: *Hunger, Thirst, Resting comfort, Ease of movement, Injuries, Disease, Human-animal bond* and *Emotional state.* A total of 54 dairy cattle experts (RES = 15%, CONS = 22%, VET = 35%, AWC = 28%) and 34 swine experts (RES = 24%, CONS = 35%, AWC = 41%) participated. Between—and within—group differences in the prioritization of criteria were assessed. AWC cattle experts differed consistently from the other cattle expert groups but only significantly for the criteria *Hunger* (*p* = 0.04), and tendencies towards significance within the criteria *Thirst* (*p* = 0.06). No significant differences were found between expert groups among swine experts. Inter-expert differences were more pronounced for both species. The results highlight the challenges of using expert weightings in aggregated welfare assessment models, as the choice of expert affiliation may play a confounding role in the final aggregation due to different prioritization of criteria.

## 1. Introduction

Animal welfare is a multifaceted concept that be described by three different ethical concerns: *biological functioning* (i.e., health and production outcomes), *affective state* (i.e., positive and negative experiences and feelings) and *natural living* (i.e., performing natural behaviours) [1]. It is often argued that different stakeholder groups attach different levels of importance to these. Thus, animal welfare viewed by the primary caretakers (e.g., farmers, veterinarians)—the primary production sector—is often thought to be described by the animal’s abilities to cope with their environment in terms of health and productivity [2,3,4,5]. However, consumers and citizens at the end of the supply chain tend to emphasize the naturalness of the animals [5,6,7] or as described by the animal’s ability to live according to its nature or *telos* [8]. Finally, the feelings-based attitude is mainly represented by scientific community (e.g., as in the welfare assessment project Welfare Quality^®^, [9,10]). This approach is a hedonistic understanding of animal welfare centred on the animal’s experiences in terms of prevention of pain and suffering and maximising positive emotions. However, it is one thing to characterise theoretically animal welfare definitions and another to understand how they come together in practice. Ethical and political views on animal welfare have given rise to a strong debate on welfare concerns [11,12,13]. Consumer behaviours do not necessarily reflect their concerns for animal welfare [12,14], but this does not prevent a strong debate based on ethical and political views [11,12,13]. As mentioned previously, the overall ethical welfare concerns are often described as being linked to professions, but this concept represents a rather categorical perception since increased awareness and political incentives might alter peoples’ perception of animal welfare and thus also challenge other stakeholder groups in their views on animal welfare. A recent study [15] also questions this postulation by reporting greater inter individual differences than between group differences in different stakeholder groups emphasizing that professional rather than educational background influences experts’ views on certain animal welfare aspects.

The choice of welfare measures and the way these are aggregated is by necessity affected by the underlying welfare understanding and the different goals of assessing welfare. Goals that may vary from certifying or classifying the level of animal welfare on individual farms, evaluating production systems across different farms to assisting the individual farmer to identify, prevent and solve welfare problems on the farm [16]. These aggregated welfare constructs are often results of expert consultations to estimate the importance of the species specific measures to be used. Caution should therefore be used when drawing conclusions on these welfare outcomes as they may be affected by the ethical considerations of the experts in the expert panels [17,18]. In Denmark, animal welfare has been highly ranked on the political agenda and several projects have been initiated on this behalf. Recently, in 2013–2016, the Danish Veterinary and Food Administration commissioned the two Danish collaborating universities to develop a national animal welfare index (DAWIN) for dairy cattle and pigs. For the construction of these indices, expert opinion was used to generate animal welfare measure weights for each species. Expert groups invited represented stakeholder groups in order to uncover a wider range of views on animal welfare, similar to a study performed in cattle [19] and swine [20] in 2012. Hence, in the two different projects giving rise to data used for the present study, similar experts groups were targeted, such as veterinary practitioners, animal welfare scientists, industry representatives involved in animal welfare activities, animal welfare control officers as well as animal protection organization officials were invited for the two expert surveys in 2012 and 2016.

The present study set out to investigate whether the professional affiliation of experts influenced their view on animal welfare measurements focusing on animal welfare assessment in sow and dairy herds—both intended for index calculation. The analysis focuses on potential between group differences but also inter-group discrepancies. As a second objective, possible changes in expert group opinions over time were assessed by a comparison with the results of a previous project running in 2009–2013 including the same expert groups but different individual experts.

## 2. Materials and Methods

The questionnaire studies were carried out at two different occasions, namely as parts of two different projects aiming at developing animal welfare indices between and at herd (carried out in 2011 and 2012) and at national level (carried out in 2016) for dairy cows and sows. Expert opinion was needed in both studies for the aggregation process of welfare measures into a single index score.

### 2.1. Expert Survey 2011 and 2012

In total, 32 dairy cattle experts were invited to an online survey in November 2011 and 14 swine experts were invited to a similar online survey in January 2012. These activities were both intended for providing weights for index calculations aiming at answering research questions of assessing and comparing animal welfare by means of two different indices, one based on existing register data and another index based on-farm collected animal based measures as described in previous studies [19,20,21]. Experts were identified as veterinarians working as either bovine or swine herd health practioners categorized into junior (<five years of experience) and senior (≥five years of experience) practioners (VET), animal scientists working as either bovine or swine production consultants and industry representatives involved in welfare activities (CONS), animal welfare control officers together with animal protection organization officials (AWC) and researchers (RES) within the field of animal welfare. A participation rate of 20 from the cattle industry and 12 from the swine industry yielded a response rate of 62.5% and 85.7% (Table 1).

For both species, the survey began by asking experts for weights for the 13 cattle and 28 sow welfare measures included in the given protocols [19,21]. At the end of the questionnaires they were asked to assign 120 points at will, i.e., minimum 0 and maximum of 120 points per criteria, between the twelve criteria *Hunger, Thirst, Resting comfort, Thermal comfort, Ease of movement, Injuries, Disease, Management induced pain, Social behavior, Species-specific behavior, Human- animal bond* and *Emotional state*.

### 2.2. Expert Survey 2016

The project regarding the animal welfare index at the national level involved a different recruitment strategy than in the previous survey. This project was commissioned by the Danish Veterinary and Food Administration (DVFA) to calculate national animal welfare indices. Four different expert groups were included, namely, practicing veterinarians (VET), animal welfare control (AWC) officers, production advisors/consultants (CONS) and researchers (RES). All four professions were recruited through a top-down approach, asking the overarching authority within each affiliation group to provide a minimum of 10 and maximum of 15 experts. Practicing veterinarians (VET) were appointed by the Danish Veterinary Associations’ subdivisions for cattle and swine; AWC officers by The Veterinary Inspection Task Force under the DVFA; CONS by the industry agency SEGES and RES by researcher in the Section for Animal Welfare and Disease Control at the University of Copenhagen [22].

Experts were contacted by email in March 2016 and at the end of the data collection responses of a total of 39 dairy cattle and 29 swine experts were received (Table 1). None of the participating experts had participated in the survey in the previous project. As in in the previous study, experts were again dealing with individual measures as well as criteria for animal welfare within the given species. In this instance experts had to distribute 100 points across the included criteria (since the DAWIN is a reduced version of Welfare Quality^®^ not all 12 criteria were included). For cattle, eight criteria were assessed: *Hunger, Thirst, Resting comfort, Ease of movement, Injuries, Disease, Human- animal bond* and *Emotional state*. Criteria for swine were slightly different, as the ten criteria for assessment were: *Hunger, Thirst, Resting comfort, Thermal comfort, Ease of movement, Injuries, Disease, Management induced pain, Social behavior,* and *Emotional state*.

### 2.3. Data Management and Weight Calculations

The outcome measures were median weights assigned to the given criteria in each of the four surveys, where all twelve criteria were rated in 2011/2012 for dairy cows and sows versus the eight for cows and ten criteria for sows in 2016. Due to the differences in numbers of criteria included and total points distributed between the species specific surveys; all weights had to be transformed into relative weights for the comparison and analyses purposes. Hence, the present study chose to include the eight overlapping criteria for each species.

Data transformation was done by first subtracting the points assigned by experts to the four additional criteria in the 2011 and 2012 and subtracting these from the initial 120 points to yield a new sum (now not fixed but relative to the individual expert) which made it possible to assign the remaining points relative to 100 points (Equation (1)):(1)$$\mathrm{Relative}\;\mathrm{criteria}\;\mathrm{weight}=\frac{\mathrm{CW}}{\mathrm{TP}-\sum \mathrm{MC}}\times 100$$
where, CW is the given criteria weight assigned by the individual expert, TP the total points assigned initially i.e., 120 and MC are the four criteria Thermal comfort, Management induced pain, Social behavior, Species-specific behavior. Finally, all criteria weights had to be harmonized, which was obtained by adding the relative criteria weights from 2011/2012 and the corresponding criteria weights from 2016 with 1/8 (Equation (2)):
(2)$$\mathrm{Final}\;\mathrm{criteria}\;\mathrm{weight}=\mathrm{CW}\times \frac{1}{8}$$

### 2.4. Data Analyses

All data management and analyses were made in R (R Development Core Team 2017, Vienna, Austria) [23]) using a general linear model with the lm function to assess the overall association between the four expert groups and their weighing of welfare criteria at a significance level of *p* < 0.05. Pairwise *t*-tests with a bonferroni correction were used to identify the individual expert groups differing significantly from another. In order to address changes in expert opinion over time between and within expert group, differences were assessed between the separate years. For the overall effect of professional affiliation on experts’ views, weights were pooled from both years.

## 3. Results

The participation rates between expert groups varied considerably between both species and surveys. As presented in Table 2, a total of 54 cattle and 34 swine experts participated. Amalgamating the two samples yielded a more balanced distribution of experts within groups; however, no swine veterinarians were available either in 2012 or 2016.

### 3.1. Prioritization of Criteria

The prioritization of welfare criteria are summarized in Table 3. Cattle experts regarded the criteria Thirst and Hunger highest and Human-animal bond and Emotional state lowest in both surveys. Disease went from a highest priority in 2011 to a less important position in 2016. A similar pattern was seen for swine expert opinion, where Hunger and Thirst ranked highest while Ease of movement and Emotional state were less prioritized in both surveys. Where cattle experts had more emphasis on the criteria Disease and Injuries measured primarily by direct outcome measures in the first survey in 2012, their focus shifted to the rather resource related criteria of Hunger and Thirst in the second survey in 2016. In contrast, swine experts assigned slightly more importance in the second survey to Disease.

### 3.2. Expert Groups

The results of discrepancies analyses for pigs are presented in Figure 1 revealing no significant effect of expert affiliation. Statistically significant discrepancies between cattle groups were found only for the criteria Hunger (*p* = 0.04), Thirst (*p* = 0.05) and Emotional state (*p* = 0.03) (Figure 2). The expert groups AWC and CONS differed significantly for Hunger (*p* = 0.04) and tended to differ for Thirst (*p* = 0.06). Surprisingly, CONS regarded Emotional state higher than the remaining three experts groups. Where RES and AWC prioritized Thirst highest, CONS and VET assigned highest weights to Disease.

Disagreement within cattle expert groups was most prominent for CONS within the criteria Hunger (SD 0.48); AWC within Thirst (SD 0.42); RES within Emotional state (SD 0.54); and VET within Resting comfort (SD 0.50). Highest disagreement within swine expert groups was found for the criteria Thirst for CONS (SD 0.76) and RES (SD 0.48) while AWC disagreed most on Emotional state (SD 0.45). Within group differences for cattle experts ranged from SD CONS = 0.23–0.47, SD AWC = 0.25–0.42, SD RES = 0.24–0.54 and SD VET = 0.26–0.62; and for pig experts SD CONS = 0.3–0.73, SD AWC = 0.19–0.45 and SD RES = 0.22–0.48.

## 4. Discussion

An interesting difference in expert participation and agreement was seen between the experts of the two species. Swine experts tended to have high agreement on the ranking of welfare criteria while cattle experts were more divided between groups. No swine veterinarians participated in either study; it is hard to say whether they could have contributed to a greater variance in the ranking, similar to the more divided picture amongst cattle experts. Among cattle experts, CONS and AWC were most often seen diverging from the other groups and in opposite directions to one another. Furthermore, no clear alignment between groups was seen across more than one single criteria. Amongst swine experts, the concordance was more pronounced, with AWC and RES being more aligned across criteria than CONS. The background of experts within the four groups could have affected the results, since AWC officers asked in the survey were predominantly veterinarians by training; RES experts could come from different backgrounds e.g., biologists, ethologists, veterinarians, animal scientist; whereas only CONS and VET had a homogenous educational backgrounds, respectively, and were also well integrated in the daily routines within the primary production. The respondents were considered as experts due their professional involvement in species-specific welfare activities—but no more direct definition on what would constitute an expert was made. Other studies have proclaimed a more stringent definition such as scientific degree (e.g., animal science, ethology or veterinary) [17,24,25] while expert weights within the WQ^®^ aggregation was derived from individuals ”*having a professional interest in animal welfare*” [5].

Previous studies could detect differences in the perspectives of affiliation groups of experts for dairy calf welfare [17], where both expert role/profession and their educational background (ethology or veterinary) had a significant influence on their welfare scores on housing factors, but not for potential welfare hazards. These results [17] are in agreement with results from another study [15] investigating the influence of expert education and current profession in regards to their opinion on the validity of welfare measures. Results involving 196 European experts showed that the current profession was more pivotal for their approach to welfare measures and the underlying criteria than their initial educational background.

The present study used expert opinion to determine welfare measurement importance for aggregation into an animal welfare index. We have addressed a relatively wide range of potential expert panel members—varying from animal welfare scientists to practising veterinarians, production advisors and animal welfare organisation delegates. This was done in order to account for the probable opinion differences. Among others, several studies [17,26,27,28] used a similar approach of balanced composition of panel participants. These authors all agree that it is crucial that experts are explicitly defined—but also in order to be able to study the differences in opinion of different stakeholders. However, it has been argued [29] that welfare scientists including ethologists and veterinarians presumably are better qualified than lay people to make judgements on the overall animal welfare state whenever the welfare judgement is to be based on a complex dataset on various welfare indicators. The findings of larger SD ranges for cattle experts belonging to the groups VET and RES in the present study are not aligned with the previous statement by [29]. However, AWC experts for both pigs and cattle, who predominantly were official veterinarians, showed smallest within-group disagreement.

Other studies have chosen a highly specialised expert approach over a heterogeneous expert panel, e.g., one study [30] using animal behaviour scientists to evaluate the welfare impact of lameness in pigs in one part and practising veterinarians to evaluate the production consequences of lameness. Another example is Welfare Quality^®^ [31] where the experts consulted were Welfare Quality^®^ project partners from the work package dealing with developing the respective measurements in as much as it was ‘assumed that they were the ones who knew the most about these measures, how they would be collected on farms, and the interpretation of the data obtained’. An obvious disadvantage of this approach seems to be that experts may by biased when having to evaluate their own contributions to the animal welfare assessment system. Another consequence of this decision was that the expert panels consisted of relatively few people (approximately five) [31], which raises another question: How many experts are needed?

The number of experts needed should be seen in relation to the characteristics of the panel. Hence, in previous studies using highly specified expert panels numbers as different as five [31], six, eight [30] and thirteen [29] have been used while numbers in studies with more heterogeneous expert panels [17,25,26,28] vary from 22 to 56. Surely a highly specified expert panel calls for less participants than a more heterogeneous one solely because a lower in between expert variation is expected. However, it can be deduced that even Welfare Quality^®^ concluded that more than five experts are needed [31] in as much as it reported that ‘it will be possible to involve more experts and recalibrate the construction’. In the present study, rather heterogeneous expert panels were targeted. This was done in order to account for the probable opinion differences between expert groups, however, the present study design does not enable conclusions to be generalised as consumers have not been included as an expert group. In order to further strengthen results from expert panels it has been advocated that ongoing inter- and intra-observer reliability testing and reporting over time is needed to assess the variations within expert groups. As a consequence, it is suggested to increase the pool of experts [32].

Comparing our findings to the WQ^®^ expert derived criteria weights, differences in the ranking of weights are evident. Even though the first expert survey was performed in 2011, Danish experts showed a markedly different prioritization compared to the, at the time, well acknowledged, but yet not that well disseminated WQ^®^ protocol. In the WQ^®^ criteria for dairy cows are ranked: *Thirst 0.27 > Disease 0.24 > Emotional State 0.17 > Comfort around Resting 0.15 > Management induced Pain 0.13 > Hunger/Ease of Movement/Human-animal bond 0.12 > Thermal Comfort/Injuries 0.11 > Social Behaviour 0.10 > Other Behaviours 0.07* [33]. In contrast, in 2011 the Danish experts still assigned most importance to *Disease*, however, *Emotional State* was regarded as a minor aspect in regards to dairy cow welfare. The only change to the following survey in 2016 was for *Disease* having a more mediocre role than earlier. One reason for the discrepancies between the WQ and the Danish experts can be found in the extensive legislation on the keeping of production animals. In 2010, the first act on the keeping of dairy cows and their offspring was ratified, defining not only minimum standards for resource—but also management-based welfare parameters e.g., mandatory claw trimming. Furthermore, herd health contracts between farmers and herd veterinarians became mandatory for dairy herds greater than 100 adult and/or 200 young cattle and yearly welfare audits were required. Thus, an emphasis on the control and preventive incentives in the production animal sector could have contributed to a different prioritization on the national level, as many welfare criteria were now considered somewhat covered by a legislative context. The same legal framework was implemented for pigs with mandatory herd health contracts for herds above 300 sows. However, no WQ^®^ weights for welfare criteria for sows are available for comparison. For both species, the low ranking of the behavioural criteria *Human-animal bond* and *Emotional state* do not correspond perfectly with the overall notion of the scientific consensus of animal welfare from the *affective state* perspective, but is probably rather a reflection of the composition of the two expert panels with a surplus of experts with a veterinary or animal husbandry background in contrast to the expert panel from [15]. Likewise, the distribution of experts within groups for each year was not balanced, reflecting the divergent results, since a larger number of AWC officers contributed in 2016 compared to the earlier study. This together with the opposite distribution of experts in the CONS group between the two study years influences the overall ranking of criteria.

## 5. Conclusions

The findings of the present study highlight how expert panel composition can affect results significantly due to either expert education or professional background. In order to avoid bias and to enhance reliability of expert surveys, balanced group numbers and the evaluation of within group variations are essential in future studies relying on expert opinion.

## Figures and Tables

**Figure 1 animals-07-00085-f001:**
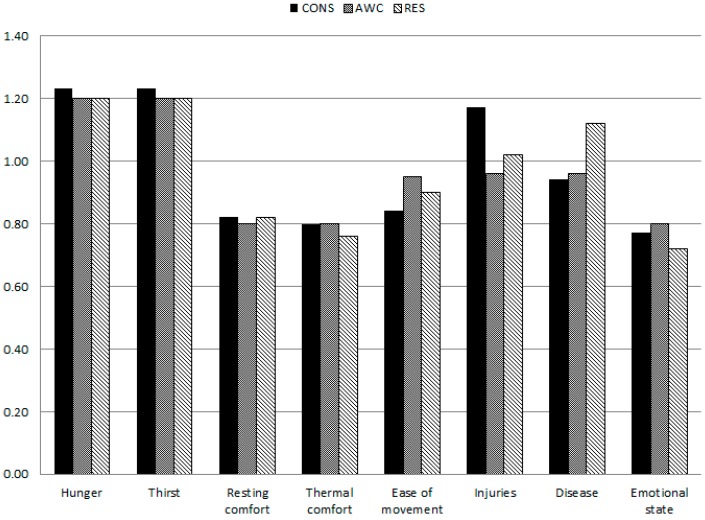
Median expert weights for welfare criteria given by three different Danish swine expert groups, production advisors/consultants (CONS), animal welfare control officers (AWC) and animal welfare researchers (RES) in two online surveys (2012, 2016).

**Figure 2 animals-07-00085-f002:**
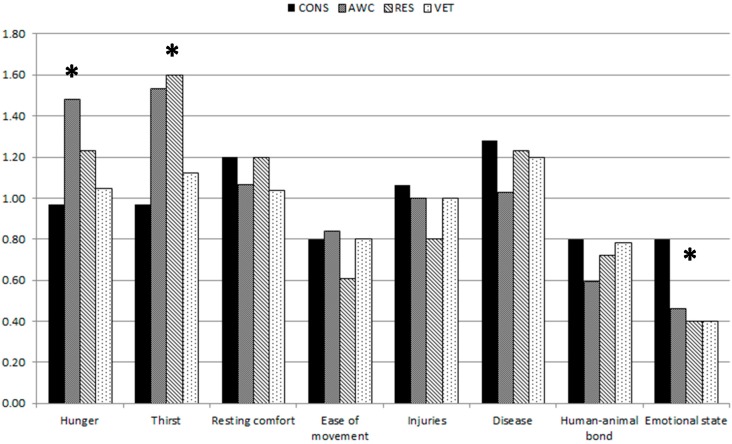
Median expert weights for welfare criteria given by four different Danish cattle expert groups, production advisors/consultants (CONS), animal welfare control officers (AWC), animal welfare researchers (RES) and bovine veterinarians (VET) in two online surveys (2011, 2016). Significant differences between groups are marked with an asterix (significance level * *p* < 0.05).

**Table 1 animals-07-00085-t001:** A summary of respondents in two online surveys on animal welfare for Danish cattle (2011) and Danish swine experts (2012).

Species		Veterinary Practice (VET)	Production Consultancy (CONS)	Animal Welfare Control (AWC)	Research (RES)	
Cattle	Invited	10	9	6	7	32
	Respondents	7	7	4	2	20
Swine	Invited	0	10	2	2	14
	Respondents	0	8	2	2	12

**Table 2 animals-07-00085-t002:** The distribution of Danish experts for cattle and swine participating in four online surveys on animal welfare in 2011, 2012 and 2016.

Affiliation	Cattle			Swine		
	2011	2016	Sum	2012	2016	Sum
Veterinary practice (VET)	7	12	19	0	0	0
Production consultancy (CONS)	7	5	12	8	4	12
Animal welfare control (AWC)	3	12	15	1	13	14
Research (RES)	2	6	8	2	6	8
Total	19	35	54	11	23	34

**Table 3 animals-07-00085-t003:** The prioritization of welfare criteria rated by Danish experts for cattle and swine participating in four online surveys on animal welfare in 2011, 2012 and 2016.

Rank	Cattle				Swine			
	2011	Median (SD)	2016	Median (SD)	2012	Median (SD)	2016	Median (SD)
1	Disease	1.3 (±0.41)	Thirst	1.6 (±0.46)	Hunger	1.20 (±0.31)	Hunger	1.2 (±0.32)
2	Thirst	1.2 (±0.24)	Hunger	1.2 (±0.46)	Thirst	1.14 (±0.75)	Thirst	1.2 (±0.44)
3	Hunger	1.2 (±0.30)	Resting comfort	1.2 (±0.51)	Injuries	1.0 (±0.51)	Injuries	1.04 (±0.27)
4	Resting comfort	1.0 (±0.28)	Disease	1.2 (±0.33)	Thermal comfort	0.94 (±0.23)	Disease	1.04 (±0.29)
5	Injuries	1.0 (±0.53)	Ease of movement	0.8 (±0.34)	Ease of movement	0.94 (±0.44)	Resting comfort	0.8 (±0.32)
6	Ease of movement	0.9 (±0.27)	Injuries	0.8 (±0.32)	Disease	0.94 (±0.48)	Thermal comfort	0.8 (±0.27)
7	Human-animal bond	0.9 (±0.35)	Human-animal bond	0.4 (±0.35)	Resting comfort	0.84 (±0.24)	Ease of movement	0.8 (±0.29)
8	Emotional state	0.5 (±0.42)	Emotional state	0.4 (±0.22)	Emotional state	0.57 (±0.32)	Emotional state	0.8 (±0.43)

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
