# Peer review of "Influence of Professional Affiliation on Expert’s View on Welfare Measures"

_animals, 2017, doi:10.3390/ani7110085_

Reviewer 1 Report

Influence of professional affiliation on expert’s view on measures

This manuscript describes results of a survey undertaken on dairy and swine welfare experts and how their professional affiliation affects their ranking of welfare issues. Overall, it is an interesting study. It could benefit from more information the professional work undertaken by the respondents as this may have been important to the outcomes.  Additionally, the outcomes need to be made clearer so that the novelty of the study can be better appreciated. The information in the discussion also needs to be related more clearly to previous studies so that it compares and contrasts the current results with previous work.

Line 28 Add space between “).” and “No…”

Line 39 Remove additional ‘,’

Line 47 Is hedonistic the correct word for this description

Line 48 Reword this sentence. I suggest removing the work ‘thing’

Line 51 Add . rather than ,

Line 51 change do to does

Line 51 Remove initial

Line 54 Change conception to concept

Line 57 Remove In and start sentence with A

Line 62 Remove ‘that’

Line 69 ‘on this behalf’?

Line 75 Remove ‘both’

Line 74 Reword sentence to improve clarity

…giving rise to the data used for the present study veterinary practioners, animal welfare scientists and industry representatives as well as animal welfare control officers and animal protection organisation officials…

Line 74 Need to make the methodology clearer here i.e. was the data from the present study collected as part of the previous study?

Line 82 were assessed not where

Line 74-83 Needs to have more clarity. The connections between the studies and where the data originated from is not clear.

Line 86 Add the between at and herd

Line 91 Might be of use to briefly outline the research questions mentioned in the other studies

Line 98 Was RES in the field of animal welfare rather than swine or dairy animal welfare

Line 99 a participation rate of 20 responses from the cattle industry and 12 from the swine industry

Line 103 Began rather than started

Line 103-107 Explain this section more. How were they asked to rate their number allocations? How many points could they give to each criteria?

Line 121 Remove of

Line 122 How many ‘previous years’?

Line 122 Change like in to as in

Line 124 Change ‘At this’ to ‘In this’

Line 129 Why no ‘Human-animal bond’ for pigs

Line 136 Might be missing something but still not sure why the eight overlapping criteria were chosen

Line 176 I’m assuming you could only compare between ones that were present in both industry questions?

Discussion

Line 196 Move discussion and conclusion heading

Line 213 Line 222 How does the background and education relate to your study? Need to relate this to your work and link them together.

Line 217 Change concordance to agreement

Line 235 Need the ‘one’ on the end?

Line 235 Change to Jensen et al ‘who’ aimed

Line 240 Needs more. What study questions were asked and how is this relevant?

Line 259 Conclusion – add s

Line 260 Have not been included

Line 262 What are the general assumptions of independence between experts? That they will have views independent of each other?

Line 264 ‘clearly report their variations as well’ Needs a full stop and needs to be reworded for clarity

Line 274 Word choice – remove playing

Overall, it seems that the discussion needs to be better related to work completed for the project.

Additional comments

I’m unsure of why the references have both the author, year and number [ ]. It’s my understanding that only the number is required for this publication?

Author Response

Dear Reviewer,

On behalf of the authors I would like to thank You for Your comments which we have tried to meet in the revision of the manuscript. In order to meet Your comments we have made the following revisions:

Reviewer #1:

This manuscript describes results of a survey undertaken on dairy and swine welfare experts and how their professional affiliation affects their ranking of welfare issues. Overall, it is an interesting study.

It could benefit from more information the professional work undertaken by the respondents as this may have been important to the outcomes.  

AU: We fully agree on this comment, however, under the given circumstances of the present study not being an initial study (2012) with a follow up in 2016, but rather amalgamated samples of expert answers these considerations were not possible.

Additionally, the outcomes need to be made clearer so that the novelty of the study can be better appreciated. The information in the discussion also needs to be related more clearly to previous studies so that it compares and contrasts the current results with previous work.

Overall, it seems that the discussion needs to be better related to work completed for the project.

AU: More results on inter-group agreement has been added both to the result and the discussion sections and the discussion has been revised in order to emphasize the current findings.

Inserted new lines 198-200: “Within group differences for cattle experts ranged from SD CONS=0.23-0.47, SD AWC= 0.25-0.42, SD RES =0.24-0.54 and SD VET=0.26-0.62; and for pig experts SD CONS=0.3-0.73, SD AWC=0.19-0.45 and SD RES=0.22-0.48.”

Deleted  old line 221: “This was also reflected  by the greater variation between individuals rather than between expert groups.

Inserted new lines 253-256: “The findings of larger SD ranges for cattle experts belonging to the groups VET and RES in the present study are not aligned with the previous statement by [29]. However, AWC experts for both pigs and cattle, who predominantly were official veterinarians, showed smallest within group disagreement. 

 Specific comments:

Line 28 Add space between “).” and “No…” – revised as suggested

Line 39 Remove additional ‘,’ – revised as suggested

Line 47 Is hedonistic the correct word for this description –

AU: It is a term often used see e.g. Appleby & Sandøe 2002

Line 48 Reword this sentence. I suggest removing the work ‘thing’ – according to reviewer #2 the sentence has been revised to: “However, it is one thing characterise to theoretically char animal welfare definitions and another how they come together in practice.”

Line 51 Add . rather than , AU: It is presumed that this refers to the revised sentence above?

Line 51 change do to does – revised to: “Consumer behaviours do not…”

Line 51 Remove initial – revised as suggested

Line 54 Change conception to concept – revised as suggested

Line 57 Remove In and start sentence with A – revised as suggested

Line 62 Remove ‘that’ – revised as suggested

Line 69 ‘on this behalf’? AU:  Unclear which sentence this comment relates to?

Line 75 Remove ‘both’ – revised as suggested

Line 74 Reword sentence to improve clarity

…giving rise to the data used for the present study veterinary practioners, animal welfare scientists and industry representatives as well as animal welfare control officers and animal protection organisation officials… 

Line 74 Need to make the methodology clearer here i.e. was the data from the present study collected as part of the previous study?

Line 74-83 Needs to have more clarity. The connections between the studies and where the data originated from is not clear.

AU:  The section Lines 74-83 (new Lines 75-81) have been revised for clarity: “Expert groups invited represented different stakeholder groups in order to uncover a wider range of views on animal welfare, similar to a study performed in cattle [19] and swine [21] in 2012. Hence, in the two different projects giving rise to data used for the present study similar experts groups were targeted as veterinary practioners, animal welfare scientists, industry representatives involved in animal welfare activities, animal welfare control officers as well as animal protection organization officials were invited for the two expert surveys in 2012 and 2016.

Line 82 were assessed not where – revised as suggested

Line 86 Add the between at and herd – revised as suggested

Line 91 Might be of use to briefly outline the research questions mentioned in the other studies

AU: Inserted new lines 98-100: “…answering research questions of assessing and comparing animal welfare by means of two different indices, one based on existing register data and another index based on-farm collected animal based measures as described in…”

Line 98 Was RES in the field of animal welfare rather than swine or dairy animal welfare                AU: Researchers were considered as experts in the field of animal welfare in general.

Line 99 a participation rate of 20 responses from the cattle industry and 12 from the swine industry – revised as suggested

Line 103 Began rather than started – revised to : “…began by asking…”

Line 103-107 Explain this section more. How were they asked to rate their number allocations? How many points could they give to each criteria?

AU: revised according to comment (new lines 113-115): “At the end of the questionnaires they were asked to assign 120 points at will, i.e. minimum 0 and maximum of 120 points per criteria, between the twelve criteria…”

Line 121 Remove of – revised as suggested

Line 122 How many ‘previous years’? – changed to “…in the previous project.”

Line 122 Change like in to as in – revised as suggested

Line 124 Change ‘At this’ to ‘In this’ – revised as suggested

Line 129 Why no ‘Human-animal bond’ for pigs

AU: The development of the sow protocol by [21] did not include avoidance distance of sows due to time

Line 136 Might be missing something but still not sure why the eight overlapping criteria were chosen

AU: The DAWIN protocol which was the subject of the expert survey in 2016 operates with only eight of the twelve WQ criteria. It is a reduced version due to time constraints in the on farm assessment, as the protocol is intended for official use and has to be carried out together with a DAWIN protocol for calves at the time of visit.

Line 176 I’m assuming you could only compare between ones that were present in both industry questions?

AU: This is merely a summary of the rankings of experts within the two species, not intended as a comparison between species.

Discussion

Line 196 Move discussion and conclusion heading

AU: Headings added at new line 212 “Discussion” and new line 312 “Conclusions”

Line 213 Line 222 How does the background and education relate to your study? Need to relate this to your work and link them together.

AU: We are unsure what the reviewer means by this question. At present the discussion consists to a large part of precisely a discussion of how education and profession affect the welfare assessment by the experts.

Line 217 Change concordance to agreement –revised as suggested

Line 235 Need the ‘one’ on the end?  - deleted “one”

Line 235 Change to Jensen et al ‘who’ aimed

Line 240 Needs more. What study questions were asked and how is this relevant?

AU: Deleted old lines 235-241 and rephrased to new lines 257-259: “Other studies have chosen a highly specialised expert approach over a heterogeneous expert panel, e.g. [30] using animal behaviour scientist to evaluate the welfare impact of lameness in pigs in one part and practising veterinarians to evaluate the production consequences of lameness.”

Line 259 Conclusion – add s – revised as suggested

Line 260 Have not been included – revised as suggested

Line 262 What are the general assumptions of independence between experts? That they will have views independent of each other?

Line 264 ‘clearly report their variations as well’ Needs a full stop and needs to be reworded for clarity

AU: Revised old lines 262-265 to new lines 287-291: “In order to further strengthen results from expert panels [32] advocates ongoing inter- and intra-observer reliability testing and reporting over time to assess the variations within expert groups.”

Line 274 Word choice – remove playing – revised to “having a more mediocre role…”

Additional comments

I’m unsure of why the references have both the author, year and number [ ]. It’s my understanding that only the number is required for this publication?

AU: All references have been edited to the required format

Reviewer 2 Report

Main comments

I commend the authors on an interesting manuscript – there are instances of animal welfare / ethics panels in many countries and these issues affect all of those. Composition and consensus are important factors in establishing guidelines and improving animal outcomes – this paper supports the need for balance across the groups and the necessity of reporting variation of opinions in the interests of transparency.

I have made some minor corrections below, but some more general points are here. I felt that the abstract might have mentioned the survey collection took part over 2 discrete time points – particularly as there was a shift in opinion between the two points and it was a stated aim of the paper. I felt the lack of a clearly distinguished “discussion” and “conclusion” section, as denoted by a subheading lead to a breakdown in the flow of the piece towards the latter half – looking at papers published in the journal it appears that those section headings are normally included, and I see no real reason for them not to be here, if only to help focus.

In the last section, line 214-221 and line 249-265 did not seem to fit as well as they might. The former section felt superfluous and I don’t think the manuscript would suffer without it as those thoughts seem well incorporated elsewhere. The latter section was an interesting aside, a key question, however it was difficult to conclude based on the very variable numbers in this study. The last two sentences are critically important though – I wonder if a shortened of this section might help to improve the flow but not lose the impact. 

Minor comments

Abstract

Line 28 insert space after bracket, before “No”

Introduction

Line 36 Delete “be said to” as repetition of “described as” in my opinion

Line 39 Delete comma after full stop (after “behaviours)”)

Line 44 Insert space between (1993) and [8]

Line 48 slightly awkward language – maybe re-word “However, it is one thing to theoretically characterise animal welfare definitions and another how they come together in practice.”

Line 51 “The consumers’ purchasing behaviour does not necessarily reflect” or “The consumer’s purchasing behaviour does not”- either single or plural consumer/s needs the “does not” rather than “do not”. “Consumer behaviours do not necessarily reflect” works too

Line 52 [14,12] – reverse order? Likewise I assume [11-13] on previous line includes 12.

Methods

Line 127 “slightly different, a the ten criteria” – delete the “a”?

Results

Line 159 line “no swine….nor 2016” is a little awkward, a lot of negatives in one sentence. “No swine veterinarians were available/included (better than eligible which implies you disqualified them?) either in 2012 or 2016”

Line 170 middle of the sentence is a little disjointed – “…shifted from emphasis on the criteria disease and injuries measured by direct outcomes to the rather resource related criteria of Hunger and Thirst”.

Line 199 is it easier to say “no swine veterinarians” as you mention earlier that invitations were issued

Line 217 “these findings” – does the “these” refer to the current manuscript or results from [17]?

Line 253 “because a less in between” – delete “a” or change to “because a lower in between…”

Line 272 insert space after “0.07”

References

Line 319 embolden “2014” to be consistent

Author Response

Dear Reviewer,

On behalf of the co-authors I would like to thank Ypu for Your helpful comments, which we have tried to meet in the revision of the mansucript by implementing the following changes:

Main comments

I commend the authors on an interesting manuscript – there are instances of animal welfare / ethics panels in many countries and these issues affect all of those. Composition and consensus are important factors in establishing guidelines and improving animal outcomes – this paper supports the need for balance across the groups and the necessity of reporting variation of opinions in the interests of transparency.

I have made some minor corrections below, but some more general points are here. I felt that the abstract might have mentioned the survey collection took part over 2 discrete time points – particularly as there was a shift in opinion between the two points and it was a stated aim of the paper.

AU: Inserted in Line 21: “At two time points (2012 and 2016) dairy cattle and swine experts from four different stakeholder groups, namely researchers…”

I felt the lack of a clearly distinguished “discussion” and “conclusion” section, as denoted by a subheading lead to a breakdown in the flow of the piece towards the latter half – looking at papers published in the journal it appears that those section headings are normally included, and I see no real reason for them not to be here, if only to help focus.

AU:  Headings added at new line 212 “Discussion” and new line 323 “Conclusions”

I

n the last section, line 214-221 and line 249-265 did not seem to fit as well as they might. The former section felt superfluous and I don’t think the manuscript would suffer without it as those thoughts seem well incorporated elsewhere.

The latter section was an interesting aside, a key question, however it was difficult to conclude based on the very variable numbers in this study. The last two sentences are critically important though – I wonder if a shortened of this section might help to improve the flow but not lose the impact. 

AU: Lines 214-221 have been kept in the manuscript while the latter section has been revised

Lines 249-256 revised to new lines 274-278: “The number of experts needed should be seen in relation to the characteristics of the panel. Hence, in previous studies using highly specified expert panels numbers as different as five [31], six, eight [30] and thirteen [29] have been used while numbers in studies with more heterogeneous expert panels [17, 25, 26, 28] vary from 22 to 56.”

Deleted lines 260-262

Minor comments

Abstract

Line 28 insert space after bracket, before “No” – revised as suggested

Introduction

Line 36 Delete “be said to” as repetition of “described as” in my opinion – revised as suggested

Line 39 Delete comma after full stop (after “behaviours)”) – revised as suggested

Line 44 Insert space between (1993) and [8] – revised according to the general format of references

Line 48 slightly awkward language – maybe re-word “However, it is one thing to theoretically characterise animal welfare definitions and another how they come together in practice.” – revised as suggested

Line 51 “The consumers’ purchasing behaviour does not necessarily reflect” or “The consumer’s purchasing behaviour does not”- either single or plural consumer/s needs the “does not” rather than “do not”. “Consumer behaviours do not necessarily reflect” works too – revised as suggested to: “ Consumer behaviours do no….”

Line 52 [14,12] – reverse order? Likewise I assume [11-13] on previous line includes 12. – revised as suggested

Methods

Line 127 “slightly different, a the ten criteria” – delete the “a”? – revised to “…slightly different, as the ten criteria…”

Results

Line 159 line “no swine….nor 2016” is a little awkward, a lot of negatives in one sentence. “No swine veterinarians were available/included (better than eligible which implies you disqualified them?) either in 2012 or 2016” – revised as suggested to: “…were available for either survey in 2012 and 2016.”

Line 170 middle of the sentence is a little disjointed – “…shifted from emphasis on the criteria disease and injuries measured by direct outcomes to the rather resource related criteria of Hunger and Thirst”. 

AU: sentence revised to: “Where cattle experts had more emphasis on  the criteria Disease and Injuries measured primarily by direct outcome measures in the first survey in 2012 their focus shifted on to the rather resource related criteria of Hunger and Thirst in the second survey in 2016”

Line 199 is it easier to say “no swine veterinarians” as you mention earlier that invitations were issued – revised as suggested

Line 217 “these findings” – does the “these” refer to the current manuscript or results from [17]? – AU: “These findings…” changed to: “Results found by [17] are in agreement…”

Line 253 “because a less in between” – delete “a” or change to “because a lower in between…”­– deleted according ti previous comment

Line 272 insert space after “0.07” – revised as suggested

References

Line 319 embolden “2014” to be consistent – revised as suggested

Reviewer 3 Report

A well written and designed study with low participation in questionnaires but providing interesting if not intuitive results. More could have been made on a comparison of year to year data if higher numbers of respondents were obtained. An inclusion of swine veterinarians would also benefit the data.

There are numerous minor grammatical errors throughout the paper, some of these are listed

Low number of responses

Line 51 – behaviours not behaviour

Line 176 statistically not statistical

Line 178-179 AWC and CONS differed significantly for …..

Line 224 relatively not relative

Line 223 on a complex dataset of various welfare indicators

Line 237 opinions not opinion

Line 246 consisted of

Line 252 than not that

Line 256 on the number of participants needed

Line 259 conclusions not conclusion

Author Response

Dear Reviewer,

On behalf of the authors I would like to thank You for Your comments which we have tried to meet in the revision of the mansucript by the following changes:

Comments and Suggestions for Authors

A well written and designed study with low participation in questionnaires but providing interesting if not intuitive results. More could have been made on a comparison of year to year data if higher numbers of respondents were obtained. An inclusion of swine veterinarians would also benefit the data.

There are numerous minor grammatical errors throughout the paper, some of these are listed

·         Low number of responses 

AU: The authors cannot find the respective annotation in the manuscript?

·         Line 51 – behaviours not behaviour – revised as suggested

·         Line 176 statistically not statistical – revised as suggested

·         Line 178-179 AWC and CONS differed significantly for …..– revised as suggested

·         Line 224 relatively not relative– revised as suggested

·         Line 223 on a complex dataset of various welfare indicators– revised as suggested

·         Line 237 opinions not opinion– revised as suggested

·         Line 246 consisted of– revised as suggested

·         Line 252 than not that– revised as suggested

·         Line 256 on the number of participants needed– revised as suggested

·         Line 259 conclusions not conclusion– revised as suggested

Best regards, Nina Dam Otten
